# Parallel Dysregulated Immune Response in Severe Forms of COVID-19 and Bacterial Sepsis via Single-Cell Transcriptome Sequencing

**DOI:** 10.3390/biomedicines11030778

**Published:** 2023-03-03

**Authors:** Alexis Garduno, Gustavo Sganzerla Martinez, Ali Toloue Ostadgavahi, David Kelvin, Rachael Cusack, Ignacio Martin-Loeches

**Affiliations:** 1Department of Clinical Medicine, University of Dublin, Trinity College, D08 NHY1 Dublin, Ireland; 2Department of Microbiology and Immunology, Dalhousie University, 5850 College Street, Halifax, NS B3H 4R2, Canada; 3Department of Intensive Care Medicine, St. James’s Hospital, James’s Street, D08 NHY1 Dublin, Ireland; 4Hospital Clinic, Institut D’Investigacions Biomediques August Pi i Sunyer (IDIBAPS), Universidad de Barcelona, Ciberes, 08036 Barcelona, Spain

**Keywords:** single cell transcriptomics, critically ill patients, SARS-CoV-2, sepsis, septic shock, immune silence, monocyte dysregulation, secondary infection, mild disease, peripheral blood mononuclear cells (PBMCs)

## Abstract

Critically ill COVID-19 patients start developing single respiratory organ failure that often evolves into multiorgan failure. Understanding the immune mechanisms in severe forms of an infectious disease (either critical COVID-19 or bacterial septic shock) would help to achieve a better understanding of the patient’s clinical trajectories and the success of potential therapies. We hypothesized that a dysregulated immune response manifested by the abnormal activation of innate and adaptive immunity might be present depending on the severity of the clinical presentation in both COVID-19 and bacterial sepsis. We found that critically ill COVID-19 patients demonstrated a different clinical endotype that resulted in an inflammatory dysregulation in mild forms of the disease. Mild cases (COVID-19 and bacterial non severe sepsis) showed significant differences in the expression levels of CD8 naïve T cells, CD4 naïve T cells, and CD4 memory T cells. On the other hand, in the severe forms of infection (critical COVID-19 and bacterial septic shock), patients shared immune patterns with upregulated single-cell transcriptome sequencing at the following levels: B cells, monocyte classical, CD4 and CD8 naïve T cells, and natural killers. In conclusion, we identified significant gene expression differences according to the etiology of the infection (COVID-19 or bacterial sepsis) in the mild forms; however, in the severe forms (critical COVID-19 and bacterial septic shock), patients tended to share some of the same immune profiles related to adaptive and innate immune response. Severe forms of the infections were similar independent of the etiology. Our findings might promote the implementation of co-adjuvant therapies and interventions to avoid the development of severe forms of disease that are associated with high mortality rates worldwide.

## 1. Introduction

In developed and underdeveloped countries, sepsis is defined as a life-threatening organ dysfunction event that results from a dysregulated host immune response to an invasive infection. This host systemic response is characterized by both pro-and anti-inflammatory mechanisms [1] and pronounced alterations in the peripheral immune compartment, including phenotypic alteration in myeloid cells, decreased responsiveness to stimuli, and a low expression of HLA-DR when compared to the persistent high expression in severe SARS-CoV-2 infections [2,3,4]. It is the most common immediate cause of death in hospitals [5] and, despite decades of research for more effective cures, it affects between 47 and 50 million people every year, with at least 11 million deaths documented (one death every 2.8 s) [6].

These numbers fail to elucidate the magnitude of the effect of the incidence of the new coronavirus disease 2019 (SARS-CoV-2). COVID-19, a heterogenous disease that frequently causes induced acute respiratory failure, severe hypoxemia, and multiple organ dysfunction syndrome in critically ill patients, has generated an unprecedented health crisis since it was declared in 2020 [7]. Persistent immune activation involving the AP-1/p38MAPK has been identified as a distinctive upregulatory mediator across immune subsets in COVID-19 [8]. This upregulation was linked with patient severity, as was the reduced diversity in CD8^+^ T cell populations when compared to with bacterial induced sepsis [9]. This further highlights the presence of trained immunity and persisting chromatin remodeling in myeloid and lymphoid cell populations in viral sepsis.

To date, there is no targeted treatment that can address patient heterogeneity or identify differences in bacterial sepsis, viral induced sepsis, or non-infectious critical illness. Treatment is merely supportive and separately addresses single clinical symptoms in reaction to changes in vital signs and biochemical parameters. Beginning antibiotic therapy during the first hours has been shown to significantly improve prognosis. Complication rates are estimated at 32.7%, and “failure to rescue” (FTR)—the risk of death after a treatable complication—is estimated at 6.8% only for post-surgical patients [10], making the diagnosis and treatment of sepsis even more challenging. These issues were further highlighted during the COVID-19 pandemic in which many clinical manifestations of severe SARS-CoV-2 infection paralleled bacterial sepsis given the long-term sequalae and protracted inflammatory dysregulation and organ hemostasis [11,12,13,14,15].

A comprehensive view of the host immune response to sepsis at the single-cell RNA-sequencing (scRNA-seq) level is warranted. Most studies on immunosuppression have focused on a specific type of immune cell or type of patient cohort, whether critically ill or presenting a mild form of the disease [16,17,18,19]. More importantly, there is not much research output that addresses a new major distinction existing in the clinical space: unlike in sepsis, long COVID is frequently diagnosed in mildly ill SARS-CoV-2-infected patients (i.e., those without hospital stays). Many immune landscape studies do not include these critically ill to non-severe, mild patient manifestations or the differences between bacterial and viral induced infections. While post-discharge complications from bacterial sepsis are referred to as post-sepsis syndrome and/or persistent inflammation, immunosuppression, and catabolism syndrome (PICS), post-discharge complications in SARS-CoV-2-infected patients are known as “long COVID” [20,21].

Deaths by COVID-19 are cases of viral sepsis resulting from an infection by SARS-CoV-2, which has arisen abruptly while traditional bacterial sepsis remains in a steady state without real ups or downs in its yearly incidence. COVID-19 is a new cause of sepsis that had already added more than 2.5 million cases [22] during the first four months of the year 2020, resulting in 0.2 million deaths worldwide. It continues to raise a threat as new pathogens arise, and antibiotic resistances emerge. Despite decades of research for more effective cures, the incidence of sepsis in the general population (400/100,000) and in ICU inpatients (25%), as well as its mortality rates (30%), have remained essentially unchanged. Survivors can develop morbid events and post-acute sequelae such as immune disturbances [23], severe recurrent infections [24], cognitive impairment [25], cardiovascular complications [26], organ dysfunction [27], and the newly termed “post-sepsis syndrome” [28]. In the search for an early diagnosis to fulfil one of the most important prognostic factors, the early administration of antibiotics, several scales are used to identify patients with sepsis. A new consensus [29] definition of sepsis and septic shock (“Sepsis-3”) was reached based on systemic indices and scores such as the SOFA score, lactate plasma concentration, and mean arterial pressure (MAP). The measurement of changes in the SOFA score has been shown to have a strong association with the 28-day mortality in patients with sepsis and disseminated intravascular coagulation [29]. However, this new definition of sepsis fails to provide a solid basis for its timely diagnosis because it requires the patient to be in organ dysfunction to confirm the diagnosis, while changes on the SOFA scale become predictive of an outcome after the patient has already been diagnosed with sepsis. It may also vary in SARS-CoV-2-infected patients and those patients that progress to septic shock. It is also important to note that there are current limitations when delineating the underlying pathophysiological mechanisms of multiorgan injury in COVID-19 that may be partly unique to severe acute respiratory syndrome coronavirus 2 (direct viral toxicity) and partly common with bacterial sepsis. A considerable proportion of patients with COVID-19 meet the requirements for a diagnosis of Sepsis-3, but it’s true prevalence has been underestimated in previous clinical reports [28].

The cellular and metabolic abnormalities caused by the disease are not specified in the previous literature. This is reflected in the fact that the current stratification criteria and the assessment of patient trajectories are purely based on the systemic parameters used by these very definitions [30]. The advent of next-generation sequencing (NGS) technologies has revolutionized the field of genomics, enabling the fast and cost-effective generation of genome-scale sequence data with exquisite resolution and accuracy and without prior genome sequence information [31]. Therefore, single-cell data analyses that are performed differently at the single-cell level compared with the cell population level can provide a clinical added value. This method can effectively identify transcriptomic differences that reflect immunosuppression and low-grade inflammation trajectories between patients with bacterial versus viral sepsis [32,33,34], with specific consideration given to the level of expression of cytotoxic genes in those critically ill, severe manifestations and the overexpression of genes [35] involved during sepsis that cause the efficient clearance of invading pathogens. These patient cohort differences at the single-cell level can guide the use of immunomodulatory therapy [36], addressing both the scale of severity and the progression of disease by displaying specific functional diversity and the downregulation of effector functions [37,38].

To further investigate these transcriptional differences, this study aimed to (1) utilize scRNA sequencing to determine dynamic changes in immune cell composition via isolated peripheral blood mononuclear cells; (2) determine the activation of interferon-stimulated gene signatures in critical and mild forms of bacterial and viral induced sepsis; and (3) identify persistent dysfunction in hyperinflammatory and immunosuppressive monocytes and distinct features of NK cells in critically ill patients with COVID-19 and septic shock. Furthermore, this study aimed to address the impact of trained immunity programs and in what way gene signatures are consistent with these functional pathways. In doing so, this study aimed to provide further improved clinical outcomes and treatment if new pharmaceutical agents account for the relationship with disease severity in both bacterial sepsis and COVID-19.

## 2. Materials and Methods

### 2.1. Analysis of Single-Cell Data

To elucidate pathways in peripheral immune cells that might lead to dysregulated inflammatory responses or protective immunity in mild versus severe COVID-19 and bacterial induced sepsis, we applied scRNA-seq to profile peripheral blood mononuclear cells (PBMCs) from sixteen patients hospitalized at St. James Hospital in Dublin, Ireland. Recruitment included four critically ill patients with COVID-19, defined as viral sepsis, who had acute respiratory distress syndrome (ARDS) and were admitted to the intensive care unit (ICU); four patients who were diagnosed with a mild syndrome of COVID-19 and were not hospitalized; four patients with non-severe bacterial sepsis; and four patients with septic shock, who were admitted to the ICU with a with poor prognosis.

### 2.2. Subject Characteristics and Specimen Collection

We collected peripheral blood from the sixteen patients enrolled in the “Sepsis Immunosuppression in Critically Ill Patients” study. After obtaining written informed consent from patients or assent from their next-of-kin, the study included a prospective, observational cohort from November 2020 to February 2021. An Institutional Research Board approval was granted by the SJH/TUH Joint Research Ethics Committee and The Health Research Consent Declaration Committee (HRCDC) under the register REC: 2020-05 List 17 on 2 March 2020 and Project ID 0428.

The eligibility criteria for the patients with severe COVID-19 included an age ≥18 years, a (+) SARS-CoV-2 nasopharyngeal swab obtained by RT-PCR, and a status of respiratory failure requiring ventilation. Patients with SARS-CoV-2 who were further phenotyped for ARDS in the intensive care unit were determined using the Berlin criteria: an acute onset of hypoxemic respiratory failure with a PaO_2_/FIO_2_ ratio (i.e., the ratio of the partial pressure of arterial oxygen to the percentage of inspired oxygen) of <300 on at least 5 cm of positive end-expiratory pressure and bilateral infiltrates on a chest X-ray [39]. Patients with a mild case of SARS-CoV-2 disease had a (+) SARS-CoV-2 qRT-PCR test, showed signs of a mild lower respiratory tract disease with an oxygen saturation (SpO2) of 94–96%, and did not require ward admission or vasopressor administration.

The screening for septic adult patients fulfilling the SEPSIS-3 criteria included patients with a sepsis diagnosis within the previous 36 h, according to the current SEPSIS-3 definition (suspected/proven infection and an increase of Sequential Organ Failure Assessment (SOFA) score by two points or more) [40]. Patients were defined as being in septic shock if they were admitted to the ICU due to bacterial infection and required a vasopressor to maintain a mean arterial pressure of ≥65 mmHg or greater and, despite sufficient fluid resuscitation, a serum lactate level of >2 mmol/L. A CV-SOFA score of 4 points reflected the Third International Consensus Definitions for Sepsis and Septic Shock [41,42]. The exclusion criteria were the following: hematological malignancy or a significant history of bone marrow disease, any immunosuppressive drugs, bone marrow or solid organ transplant recipients, and leucopenia (<1000 mm^−3^). Further clinical characteristics revealed a significant difference in the overall hospital length of stay and an increase in the ICU length of stay in the septic shock patients when compared to the severe COVID-19 and sepsis group. Overall, the in-hospital mortality was linear across both septic patients and severe COVID-19 patients. It was noted that the severe COVID-19 group had a significantly higher percentage of continuous renal replacement therapy (CRRT) when compared to the bacterial induced sepsis and septic shock group; refer to (Table 1).

Sepsis patients were managed using sepsis resuscitation bundles that were embedded within the electronic medical record system order sets and clinical notes. Briefly, sepsis patients underwent early, goal-directed fluid resuscitation, received empiric antibiotic therapy tailored to the presumed anatomic site of infection, and were referred for early source-control procedures when appropriate. Patients with a SARS-CoV-2 infection who were adjudicated as being septic (i.e., demonstrated evidence of life-threatening organ dysfunction due to infection) per the Sepsis-3 guidelines [40] received similar care with sepsis management bundles, though with the caveat that intravenous fluid resuscitation could be limited to less than the standard, initial 30 mL/kg of intravenous fluid bolus, especially for patients with right heart failure, acute kidney injury (AKI), or who were at a high risk for hydrostatic-pulmonary-edema-related respiratory failure.

The failures in different organs represent the interplay of the immune system within organ crosstalk. Recently published observational studies highlighted the interplay of AKI, immune dysfunction mechanisms in critical illness, and their link to a hyper-inflammatory state. Both impairment and activation pathways are shown to be involved in immune natural cells in hemodialysis (HD) patients that also have active infections. These studies reported a need for a discrete method for measuring adaptive immune system dysregulation and identifying a distinct inflammatory endotype that reflects reduced defense mechanisms such as phagocytic capabilities or the impairment of the antigen presentation function [43,44]. This distinct immune dysfunction profile was observed in our cohort of patients who were critically ill with severe COVID-19: when compared to the sepsis and septic shock patients, the patients with COVID-19 demonstrated a weakened CD4 T cell response and a reduction in the naïve cell response after CRRT.

### 2.3. Details of the Step-by-Step Method

For all critically ill and non-severe patients with viral and bacterial induced sepsis, blood was collected into 10 mL EDTA tubes (Becton, Dickinson and Co., Franklin Lakes, NJ, USA) and loaded using SepMate 50 mL tubes (Stemcell Technologies, Saint-Egreve, France). Gradients were centrifuged at 1200× *g* for 10 min with the brake on at room temperature. The cell interface was carefully removed by pipetting and washed with PBS-EDTA by centrifugation at 400× *g* for 7 min. PBMC pellets were suspended in an ammonium chloride solution (Stemcell Technologies, France) and incubated for 10 min at room temperature on a mixing platform to lyse contaminating red blood cells. The lysed pellet was resuspended in 10 mL of PBS (a small sample was taken for counting) and underwent spin suspension at 400× *g* for 7 min with the brake on. Isolated PBMCs were finally washed with PBS-EDTA and then resuspended for downstream analyses. The blood was processed within 4 h of collection for all samples. All samples obtained on the day were processed side-by-side to avoid variation from processing. They were analyzed fresh on the BD Rhapsody Single-Cell Analysis System platform to avoid clumpy cells and achieve a higher viability % of cells.

### 2.4. scRNA Sequencing by Seq-Well

The expression of 399 transcripts relative to the immune cell system (Immune Response Panel HS, cat. 633750, BD, Biosciences, Franklin Lakes, NJ, USA) was obtained at a single-cell level on human PBMCs using the BD Rhapsody Single-Cell Analysis System platform (BD, Biosciences, Franklin Lakes, NJ, USA). Specifically, human mononuclear cells were stained with FITC-conjugated antibodies (*CD19*, *CD14*, *CD16*, *CD3*, *CD4*, *CD8*, *CD304*, *CDTCRgd*, *CD56*, and *HLA-DR*), refer to (Figure 1).

BD Rhapsody cartridges were each loaded with 10,000 pooled cells derived from each subject for single-cell separation. There was a total of sixteen cartridges. The single cells were isolated using single-cell capture and cDNA synthesis with the BD Rhapsody Express Single-Cell Analysis System, according to the manufacturer’s recommendations (BD Biosciences, Franklin Lakes, NJ, USA). The 399 targeted transcripts of the BD Rhapsody immune response panel were amplified with the BD Rhapsody targeted amplification kit (cat. 633,774, BD, Biosciences, Franklin Lakes, NJ, USA), according to the manufacturer’s instructions. Unwanted PCR products and other small molecules were excluded by performing a side cleanup with AMPure XP Beckman magnetic beads (cat. #A63880, Beckman Coulter, Brea, CA, USA). Quantity and quality control of the DNA were performed using the Qubit^TM^ dsDNA HS Assay Kit (cat. # Q32851, ThermoFisher Scientific, Waltham, MA, USA) and the electrophoresis system Agilent 2200 TapeStation, cartridge (cat. #5067–5584, Agilent, Santa Clara, CA, USA).

### 2.5. scRNA-seq Computational Pipelines and Analysis

Cell-expressing data matrices were obtained from the single-cell sequenced samples from BD Rhapsody. Unique molecular identifier (UMI) filtered reads were obtained with the application of the distribution-based error correction (DBEC) algorithm. Patients from each medical condition were grouped together. The common cell types of each condition were obtained with the merge function in base R (version 4.1.2). All patients of each condition were combined into a single cell-expressing matrix. Genes and cells with low coverage were excluded, retaining the top ten most-expressed genes across nine distinct immune cells from patients with three diagnoses. The exclusion criteria considered the measurements of the loading scores of a principal component analysis (PCA) object, which was performed using the stats package in R and considered the first two components to explain the variance of the data. We built heatmaps representing the pattern of expression of genes expressed by cell types that were shared by at least two medical conditions. The transcript counts were scaled through the scale function in R. Datasets were then elongated with the melt function, and the visualization of heatmaps was achieved using the geom_tile function of a ggplot2 object.

A low dimensional analysis of immune cells expressing genes was performed to group cells by their transcripts. A t-distributed stochastic neighbor embedding (T-SNE) analysis was conducted using the RTsne package (version 0.16) in R. Perplexity values of 5, 30, and 50 were tested as reported in Wattenberg et al. (2016) [45].

**Figure 1 biomedicines-11-00778-f001:**
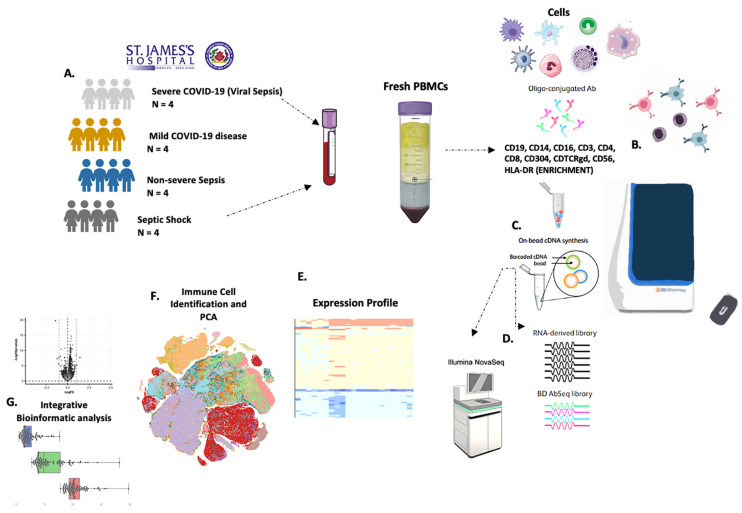
Single-cell transcriptional landscape of severe and non-severe patients with viral and bacterial sepsis and immune cell dysfunction. (**A**) A schematic outline depicting the experimental workflow for data collection from the published literature and the subsequent integrated analysis. Numbers indicate the number of samples of different cohorts (severe COVID-19, mild COVID-19, non-severe sepsis, and septic shock) and expression of 399 transcripts analyzed. (**B**) Specifically, human mononuclear cells were stained with FITC-conjugated antibodies (*CD19*, *CD14*, *CD16*, *CD3*, *CD4*, *CD8*, *CD304*, *CDTCRgd*, *CD56*, and *HLA-DR*). (**C**) Single cells isolated using single-cell capture and cDNA synthesis with the BD Rhapsody Express Single-Cell Analysis System, according to the manufacturer’s recommendations. (**D**) The BD AbSeq workflow integrated into the BD Rhapsody system. (**E**–**G**) Cells are colored by cell subtypes. Dashed circles indicate the major cell types; differential expression profiles were run among mild and severe manifestations of infection and have varied immune cell profiling.

### 2.6. Functional Analysis

Lists of the unique, most commonly expressed genes by cell type across progressing disease models were submitted to the Protein Analysis Through Evolutionary Relationship (PANTHER) tool (version 17, 22 February 2022). The results regarding pathways, biological processes, and molecular functions were converted into pie and bar charts.

## 3. Results

In order to test our hypothesis, i.e., that hyperinflammatory monocytes and immunosuppressive monocytes may be derived from the abnormal activation of normally activated monocytes in response to SARS-CoV-2 infection and introduce novel, substantial phenotypic differences between severe and non-severe patients with viral and bacterial sepsis, we (i) clinically characterized our cohort; (ii) captured differences in patterns of immune gene expression among COVID and non-COVID sepsis-induced patients; and (iii) compared the single-cell profiling of patients with organ disfunction over the course of the manifestation of their dysregulated immune response (i.e., mild and severe).

### 3.1. Differential Expression Profiles in Bacterial and Viral-Induced Sepsis

To verify if viral- and bacterial-sepsis-induced patients are clinically different in their immune response, we targeted the 10 genes that are most responsible for data variance in gene expression in COVID and non-COVID patients (for both severe and mild forms of the disease) in 10 distinct cell types. First, in the severe form of disease, we found significantly different expression levels of the genes S100A12, FOSB, DEFA3, CEACAM8, and CD24 (Figure 2A) from which all genes presented an upregulated profile (Figure 2B) in patients with septic shock (i.e., non-COVID). Regarding the profile of the mild form of disease, all 10 genes showed significantly different expression rates (Figure 2C), i.e., S100A12, LGALS1, IGLC3, IGHG1-secreted, IGHA1-secreted, HLA-A, FOSB, ENTPD1, DEFA3, and CD24 were found to be more expressed in non-COVID patients (Figure 2D). Thus, in general terms, the expression of genes by different cell types was found to be higher in both sepsis and septic shock patients.

### 3.2. Mild and Severe Manifestations of Infections Have Varied Immune Cell Profiling

We profiled the single-cell gene expression of mild and critically ill patients separately in order to depict clinical differences/similarities with respect to immune response. First, we depicted different profiles in mild cases of COVID and bacterial sepsis. In (Figure 3A), we show that the gene expression level of 10 immune cells is distinct in COVID and non-COVID patients (bacterial sepsis/septic shock) as there is little overlap. Next, in (Figure 3B), we isolated each cell’s most expressive gene. We report a significance in the expression levels of DEFA3 by CD8 naïve T cells and the FOSB gene in CD4 naïve T cells, CD4 memory T cells, and natural killers. The expression of the gene S100A12 was found to be significantly different in both classical and non-classical monocyte cells. Dendritic cells expressed differentiated levels of the ENTPD1 gene and, finally, the expression of IGLC3 by B cells also demonstrated significant differences. In (Figure 3C), we show the higher immune cell counts for non-COVID patients, especially B cells, classical monocytes, CD4 and CD8 naïve T cells, and natural killers. Moreover, we report similarities in severe cases of COVID and sepsis (i.e., septic shock). First, in (Figure 3D), the immune cells expressing 400 genes overlap among COVID and non-COVID patients. Next, in Figure 3E, we isolated the gene of each cell type according to their contribution to the data variance. Statistically significant differences were found in CD8 naïve T cells expressing the DEFA3 gene, CD4 naïve T cells expressing FOSB, and both classical and non-classical monocytes expressing S100A12. Finally, in Figure 3F, the cell count of both disease models portrays more balance than in mild manifestations of the diseases.

### 3.3. Functional Analysis of Genes Expressed in Different Cell Subtypes across Patients

First, we used a PCA analysis to measure the loading scores of the first component to extract the 10 most commonly expressed genes in each cell type across progressing disease models. In the PCA analysis, the loading scores of these genes contributed to an explanation of the variance of each independent dataset. We display the unique genes found in each disease model in Appendix A.

To differentiate the progressing (i.e., mild, and severe) viral and bacterial sepsis, we performed a functional analysis of the genes involved in each stage of each disease. First, the genes involved were used to calculate the enrichment in each pathway to characterize the patients. We report that the pathway for T cell activation and the CCKR signaling map were more abundant among the genes of non-severe COVID-19 patients, while patients with the severe form of the disease demonstrated genes that contributed less to the T cell activation pathway (Figure 4A,B). Similarly, T cell activation was heavily observed. However, interestingly, the pathway inflammation mediated by cytokine and chemokine signaling was distinctly upregulated in the mild COVID-19 cohort. Moreover, in patients with sepsis, we found the Parkinson disease pathway to be more abundant in the non-severe form of the disease (Figure 4C). Additionally, the genes responsible for the angiogenesis pathway were only found in patients with severe forms of sepsis (Figure 4D) (i.e., septic shock).

Through an analysis that considered biological processes to differentiate progressing viral and bacterial sepsis (Figure 5), we determined that the immune system and multicellular organismal processes were found at a higher extent in the mild COVID-19 cohort (Figure 5A) and in the non-severe sepsis cohort (Figure 5C), indicating that both viral and bacterial induced sepsis are similar with respect to their biological processes. Septic shock patients demonstrated a higher upregulation of the metabolic pathway when compared to both severe COVID-19 patients and those with a mild form of the disease.

Lastly, in Figure 6, we show the molecular functions that the most commonly expressed genes in the two progressing disease models are responsible for. First, we report less molecular functions performed by the genes involved in sepsis (Figure 6C), i.e., catalytic activity, transcription regulator activity, molecular transducer activity, and binding. We report that the catalytic activity and binding functions were more abundant in patients with severe sepsis (i.e., septic shock, Figure 6D), while the remaining functions were higher in patients with non-severe sepsis. Second, we report a higher extent of function associated with the genes of COVID-19 patients (Figure 6A,B), which include molecular function regulators, low-density lipoprotein particle receptors, molecular adaptors, and transport activities (this one was only present in patients with severe COVID-19). We report that the molecular transducer and catalytic activities were found at a higher extent in non-severe COVID-19 patients, converging with the sepsis model.

## 4. Discussion

A major goal of the present comprehensive analysis approach was to examine the variability in the transcriptional changes responsible for the immunosuppressive phase of both viral and bacterial induced sepsis, i.e., after the initial hyper-inflammatory phase. To determine the effect on the host immune response, we utilized scRNA sequencing of the key immune cells to further identify novel inflammatory gene expression patterns across these distinct cohorts. Applying our analysis to multiple, well-defined cohorts ranging from mild to severe forms of disease and distinct pathogen origins, we identified hallmarks of viral sepsis severity and specificity involving immune cell populations and their development, components of innate and adaptive immunity, and connectivity with the inflammatory response in the mild forms of disease and the septic shock cohort. Higher immune cell counts for both bacterial sepsis and septic shock patients were noted, especially in B cells, classical monocytes, CD4 and CD8 naïve T cells, and natural killers. We report significance in the expression levels of DEFA3 by CD8 naïve T cells, the FOSB gene in CD4 naïve T cells, CD4 memory T cells, and Natural killers. Notably, the dendritic cell population expressed differentiated levels of the ENTPD1 gene in bacterial induced sepsis. This is in opposition to the previous literature, which noted ENTPD1 as a key mediator of the purinergic pathway in the form of ENTPD1/CD39, an ectonucleotide enzyme that scavenges pro-inflammatory ATP and ADP into AMP in critically ill COVID-19 patients. It is linked to a poor clinical outcome and is defined by exhausted T cells [35].

Moreover, in critically ill cases of COVID and sepsis (i.e., septic shock), we report similarities that highlight a clear, pre-clinical parallel for sepsis and COVID-19 models including identical research niches of (i) mild-to-severe disease phenotypes, (ii) defined cohort targeting, and (iii) their elected pathophysiological insights (e.g., the compartmentalization of responses). A high percentage of COVID-19 patients hospitalized in the ICU me Sepsis-3 criteria and presented infection-associated organ dysfunction. The medical and scientific community should address pathogen-specific treatments such as remdesivir, which may accelerate clinical recovery, whereas immune-targeting therapies were tested through clinical and demonstrated conflicting results. Authors still focus on secondary bacterial infections in COVID-19 cohorts rather than considering viral sepsis as a form of ARDS, shock, and other organ failures which have also been associated with *influenza A*. The frequency of opportunistic infections is likely to rise after the widespread introduction of dexamethasone, posing an eminent clinical threat in addition to the rise of antimicrobial resistance.

Past research studies that assessed the immune response in patients with COVID-19 identified a “cytokine storm” downstream process occurring in patients in the first stages of the disease [32]. This theory proposed the development and implementation of anti-inflammatory drugs such as corticosteroids and IL-6 inhibitors. It is, however, surprising that a recent metanalysis found a moderate IL-6 release in the innate immune response of patients with bacterial sepsis when compared to patients with COVID-19, and septic shock, with the presence of ARDS. The highest levels of IL-6 and pyroptosis-related cytokines (IL-18 and IL-1α) were found in septic shock patients [33].

Interestingly, based on numerous studies conducted over the last two decades, anti-inflammatory agents have been tested and demonstrated to fail in patients with sepsis. Bacterial co-pathogens are commonly identified in viral respiratory infections and are important causes of morbidity and mortality in critically ill patients. The biological changes that underpin critical illness states of bacterial infection in patients infected with SARS-CoV-2 are not well understood.

Monocytes from patients with sepsis demonstrate a decreased responsiveness to stimuli and have a lower expression of human leukocyte antigen–DR (HLA-DR) when compared to patients with viral induced sepsis [46,47,48]. This finding that mirrors a recent paradigm shift that was published a decade ago in patients with sepsis. Kaufmann et al. (2018) proposed that there was an imbalance towards an anti-inflammatory response in patients with sepsis when compared to viral infections [49,50]. Regarding viral sepsis, Bermejo-Martin et al. (2021) found a bi-phasic pathway in patients with severe influenza and severe COVID-19 [51]. There was a first phase manifested by hypercytokinemia due to a non-controlled viral infection, followed by an impaired adaptive immune response at a cellular level [52].

Nonclassical monocytes were only found in septic shock patients, while CD8 memory T cells and dendritic cells were only found in patients with mild COVID and sepsis, respectively. These findings demonstrate that changes in monocyte cell surface receptor expression during septic shock leads to an ongoing immunomodulation of monocytes, specifically the upregulation of the non-classical subtype. During a viral, chronic infection there is a depletion due to altered IFN-α production [53].

Notably, our analysis revealed that monocytes from patients with a mild SARS-CoV-2 infection and non-severe sepsis showed a differential expression of calprotectin EN-RAGE (S100A12), an endogenous TLR-4 ligand. S100A12 expression increases during sepsis, and its expression is known to be released from granulocytes in vitro and in vivo by an inflammatory challenge [54]. A recent gene expression analysis confirmed that S100A12-activated monocytes induce inflammatory gene processes, thereby acting as an amplifier of innate immunity during early inflammation and the development of sepsis [54].

This may be due to a reduced interferon (IFN) response, which has been demonstrated in severe cases of COVID-19. These patients are associated with a diminished capacity for viral control. This suggests that in the case of our Gram-negative, bacterial induced sepsis patients, there is an activation of the NLRP3 inflammasome pathway involved in the increased expression of S100A12 in the pathogenesis of sepsis-induced ARDS. This was demonstrated by Han et al., (2020) with a sepsis-induced ARDS replicated using a CLP mouse model and normal and human bronchial epithelial (NHBE) cells [55]. This could provide clinically useful insights into the impact of S100A12 on *Pseudomonas* as it might have the potential for use as a therapeutic intervention in addition to antibiotics to combat future infections and serve as a useful biomarker of pulmonary injuries in clinical diagnoses of sepsis-induced ARDS [55].

This highlights a possible negative feedback mechanism following a Gram-negative, pathogen-associated, molecular-pattern-induced inflammation that can be arguably driven by an upregulated interferon-gamma-mediated signaling that is induced by SARS-CoV-2. Compositional and molecular changes across disease stages reflect an activation status of non-classical monocytes in septic shock that results from an initiated IFN-γ signaling in response to infection, suggesting that the interferon gamma response may be aberrant, with its highest degree in SARS-CoV-2-induced sepsis. This supports bacterial-sepsis-induced T cell exhaustion leading to an evident loss of inflammatory cytokine IFN-γ and TNF-α production from T cells following stimulation [56]. Previous studies observed a decrease in the production of IFN-γ from both Th1 and Th2 populations in sepsis patients [57,58]. These cells persisted in displaying significant reductions in their secretory profiles, which may explain the loss of the CD4+ T cell subset in the severe septic shock cohort.

Different researchers have published evidence that in patients with a severe presentation of COVID-19, there was an altered capacity of the innate immune control of COVID-19 infections, facilitating viral evasions. This might have important implications beyond the mechanistic observational evidence. Antivirals have shown to not be very effective in patients affected by COVID-19 infections. This problem is most likely not related to the anti-viral itself but the timing at which it is administered. A combination of awareness and prompt recognition might represent the best approach to treating these patients as the antiviral effect might be limited when the patient is already in a severe condition.

In the case of bacterial induced sepsis patients, a possible activation of the NLRP3 inflammasome pathway, which has been shown to be involved in the effect of S100A12 on the pathogenesis of sepsis-induced ARDS, was revealed. Even though ARDS may have several causes, including aspiration, major burns, or trauma, the most common cause of ARDS is sepsis. Moreover, patients with sepsis and ARDS are those affected by the highest inflammation levels. This finding is also surprising as patients with COVID-19 are usually those affected by single-organ failure. These patients develop severe hypoxemia, which seems to be a trigger mechanism for damage in other organs, while for patients with sepsis, endothelial damage and coagulation activation are the classically well-described pathophysiological mechanisms related to the development of multiorgan failure. Cell populations of classical and non-classical monocytes have been demonstrated to be deeply involved in the immunopathogenesis of both systemic and organ (lung) hyper-inflammatory manifestations of severe COVID-19 [59].

We found an upregulation of HLA-A in patients with sepsis compared to a dysregulation in patients with mild COVID-19. By exploring the temporal changes in the transcriptome profile from the bacterial induced sepsis and mild COVID patients, we found dramatic differences that predominantly involved CD14+ monocytes following sepsis recognition [60]. The CD14+ monocytes of patients with bacterial induced sepsis demonstrated an intense response to stimulation, stress, and inflammatory behaviors. Kim et al. (2010) found that in patients with a surgical insult and inflammation, HLA-DR expression was modulated differently in CD14 HIGH (classical) versus CD14 LOW (inflammatory) monocytes after systemic inflammation [48,60].

There was a B cell upregulation in patients with severe COVID-19 and sepsis. Differentially expressed genes were enriched in components of the antibody-processing (IGLC3) marker. This signature was selectively found to be upregulated in patients with severe COVID-19, reinforcing the concept that interferon signaling is increased in COVID-19 patients due to distinct cell states during acute infection and its contribution to host protection with respect to survival.

The T cell numbers, including total T cells and CD4+ and CD8+ T cells, in the severe disease group were significantly lower than those in the mild disease group. T cells play an important role in both entities, and they have been found to be decreased in patients with a severe presentation of both sepsis and COVID-19. On the other hand, lymphocyte CD4 and CD8 T cells, B cells, DCs, macrophages and monocytes, and NK cell depletion has been observed in septic patients. The clinical implications are relevant in patients with COVID-19 as many of the patients that survived the aggression of viral pneumonia ultimately developed secondary infections due to a high degree of immunoparalysis. During the last two years, the number of secondary infections has increased, and we have seen many reports showing that in patients with COVID-19, mortality was associated, after the acute phase of COVID, with the development of respiratory infections. This might represent local damage in the lung due to local immune alterations; however, some authors have also reported high rates of bacteremia at a systemic level.

This research focus, which relates to the local damage and recruitment of not only derived PBMCs but other tissue source samples, is warranted to assess local immune alterations and differences in microbiological results prior to the administration of antibiotic therapy or immunosuppressants. This study would have benefited from reference data across certain time points to capture dynamic changes of the individual immune cells. This study mainly reflected on early dysregulation endotypes at time points [T0] across bacterial induced sepsis compared to viral sepsis (COVID-19) rather than cellular reprogramming and immune dysregulation trajectories across cohorts in the ICU. Regarding the potential limiting factors of this study, we draw attention to the non-heterogeneous set of genes which yielded detectable transcripts in the scRNAseq analysis. If we had been able to detect a more heterogeneous set of genes or included a larger sample size, the interpretation of our results could have been expanded. Despite these limitations, the differences and the parallelism of distinct degrees of severity were captured well with the available information. We are also aware of the limitations regarding the single-institution design of this study, which limits reproducibility or validation against other cohorts or clinical settings.

Recent research highlighted that patients with sepsis are at risk of developing a nosocomial acquisition of multidrug-resistant pathogens. This is a major healthcare concern due to multiple antibiotic courses that will also alter the microbiome of critically ill patients [61,62]. These nosocomial infections are also linked with the function of a cell type that can play a cornerstone role in the understanding of the clinical progression of patients. During the early stages of septic shock, NK cells may play a key role in the promotion of the systemic inflammation, as suggested by mouse models. Alternatively, at a later stage, NK-cell-acquired dysfunction could favor nosocomial infections and mortality. Although specific effects of NK cells on viral infection and sepsis have not been previously elucidated, our results demonstrate that active NK cells may effectively control viral infection in the early stage by directly killing infected cells or promoting the infiltration of other immune cells, such as neutrophils, T lymphocytes, and B cells, into the lesion. They are highly present in our sepsis group due to the process that hyperactivated NK cells undergo. They are known to produce excessive levels of pro-inflammatory factors by releasing granzymes, subsequently causing organ injury and even death, which further informs the differentiated expression in sepsis.

Current protocol treatment guidelines for ICU patients with bacterial induced sepsis and COVID-19 are largely based on cohorts that share a dysregulated immune response profile but are limited in their incorporation of severity-related and mortality-related gene-sets that can be used to inform responses to treatment trajectories. Determining the underlying immune pathogenesis and cellular reprogramming across the spectrum of COVID-19 severity and bacterial induced sepsis and septic shock remains an important clinical challenge. The examination of this scRNA-seq informative model revealed monocyte dysregulation in immune signaling networks and immune silence that differentiates COVID-19 (viral sepsis) severity from sepsis and the mild COVID-19 subgroup. Therapeutically, the reported loss of TCD8 naïve cell expression in septic shock and the TCD8 memory cell in in mild forms of SARS-CoV-2 can be used as a pipeline for further drug discovery programs, specifically the advancement of immunotherapy and immunomodulation agents.

There are a number of Phase 1 (and some 2a) trials in the pipeline specifically for the implementation of allogenic NK cellular transfer, CAR-NK cells secreting an IL15 superagonist, and anakinra or rhIFNy adjunctives in COVID-19 and sepsis with hyperinflammation. However, Phase 3 trials are far from being implemented (ClinicalTrial.gov# NCT04344548, NCT04797975, NCT04324996, NCT04990232) [63]. Considering that the reduction in Tregs levels could be a reason for the overstimulation of the immune system and lung damage in severe COVID-19 patients, there is potential in enhancing the quantity and recovery function of Treg cells, such as through the expansion of antigen-specific Treg, which can regulate CD8^+^ effector T cells against viral sepsis. Monocyte dysregulation can also be addressed by having an immunomodulatory target that inhibits the proliferation and function of polarizing monocytes to anti-inflammatory M2 macrophages. This can facilitate the regeneration of damaged pulmonary epithelial cells and promote alveolar fluid clearance in patients who have severe COVID-19 with secondary infections.

## 5. Conclusions

This scRNA-seq study is unique as it aimed to delineate the differences and parallelisms of patients with different degrees of severity of sepsis and COVID-19. We found that the findings supported our initial hypothesis, and we found that there were no major differences in most cells that expressed genes in critical conditions independent of the causing pathogen. We identified specificity involving immune cell populations and their development, components of innate and adaptive immunity, and connectivity with the inflammatory response in the mild forms of disease, independent of the causing pathogen. Notably, our mild disease sepsis cohort developed a loss of immunosuppression from CD8+ T cells by naive T cells, suggesting a state of “immune silence” which can be related to the extensive immaturity of this cell population due to an abnormal and skewed myelopoiesis or a pre-existing comorbidity, making this group of patients unable to cope with the hyper-inflammatory state.

Interestingly, NK cells were also expanded in the course of the disease, although they were slightly decreased in reference to the disease severity. This which contradicts a previous study, which suggested the apoptosis of NK cells in sepsis compared to critically ill COVID-19 patients [63]. Thus, the sepsis-induced immunoparalysis state can be said to develop numerical and NK-cell-intrinsic functional impairments, supporting an instructive notion for future studies aimed at restoring NK cell immunity in bacterial induced sepsis [63,64,65,66]. This study was limiting in that it did not address the temporal change of the immune response after infection or how therapy infusions could mitigate further mortality risks or differences in the immune response. Longitudinal studies are warranted to dissect these different disease developments and to determine appropriate, NK-cell-based and other immuno-adjuvant therapies that are effective in sepsis and septic shock and may also be efficacious in other COVID-19 patient cohorts with viral induced sepsis. Hence, to determine if the host response could lead to further mortality risk and the development of severe forms of disease, there is a need for the development of immunomodulatory interventions and engineered CAR-NK cells that can harness existing immunity in pre-clinical models and clinical trials.

## Figures and Tables

**Figure 2 biomedicines-11-00778-f002:**
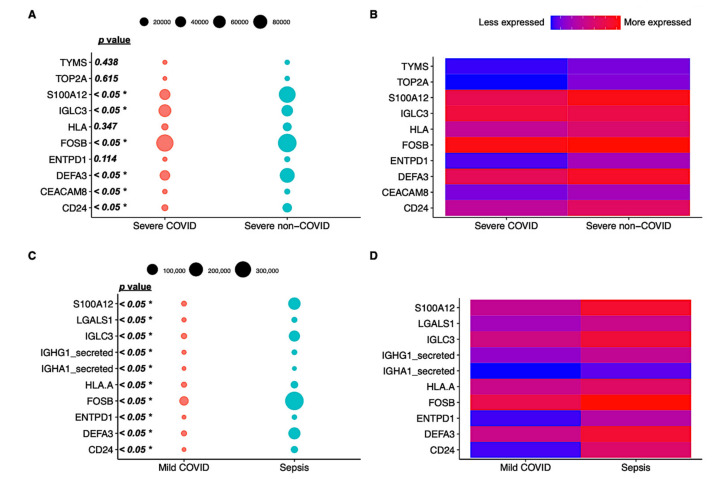
Distinctive single-cell level expression profiles of mild and severe forms of COVID-19 and bacterial sepsis. Severe forms of disease differed in expression levels of the genes S100A12, FOSB, DEFA3, CEACAM8, and CD24 (**A**) from which all genes presented an upregulated profile (**B**) in patients with septic shock (i.e., non-COVID). The mild form of disease, all 10 genes showed significantly different expression rates (**C**), i.e., *S100A12, *LGALS1, *IGLC3, *IGHG1-secreted, IGHA1-secreted, HLA-A, FOSB, ENTPD1, DEFA3, and CD24 were found to be upregulated in non-COVID patients (**D**).

**Figure 3 biomedicines-11-00778-f003:**
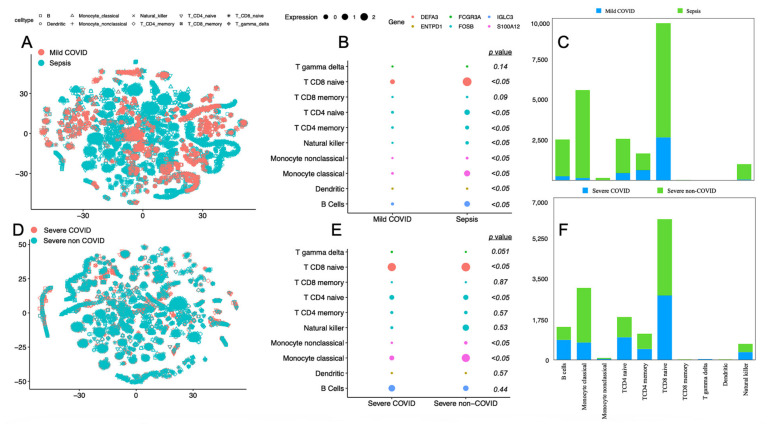
Single cell RNA-seq profile of severe and non-severe patients with viral and bacterial sepsis. We show t-SNE plots of 10 immune cells expressing 400 different genes in (**A**) mild COVID and sepsis patients and (**D**), severe COVID and septic shock patients. The dimensionality reduction achieved with t-SNE in order to bring 400 genes expression rate to two dimensions (x and y) considered a perplexity value of 30. Next, in (**B**,**E**), we, performed a Principal Component Analysis (PCA), targeted the gene that most contributes to the variance of a subset of the data composed only by one cell type. The measuring of the loading scores indicated the genes that best explain the first component of the PCA. Data distribution was assessed with the Shapiro Wilk test encompassing each disease model (COVID and non-COVID) and its severity (severe and non-severe). Then, the means of each two genes were compared either with a *t* test or Wilcox rank test. The full description of the *p* values is included in Appendix A. In (**C**,**F**) we accounted for the number of cells of each cell type identified by the single cell analysis.

**Figure 4 biomedicines-11-00778-f004:**
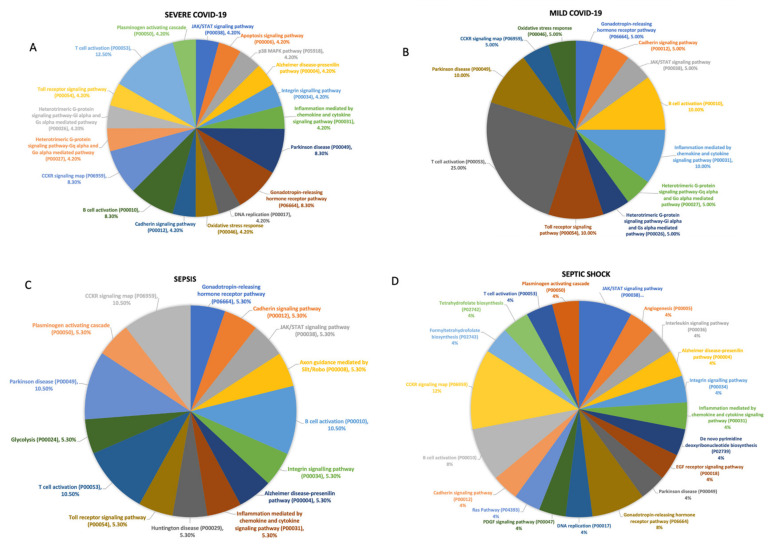
Pathway of the most commonly expressed genes in different cell subtypes across patients. Individually, the unique, most commonly expressed genes in COVID-19 and bacterial sepsis patients with mild and severe disease manifestations, found in Appendix A, were submitted to the PANTHER software. A pathway involvement analysis of the genes was returned by the software and was later coded into pie charts. T cell activation and the CCKR signaling map were more abundant among the genes of non-severe COVID-19 patients, while patients with the severe form of the disease demonstrated genes that contributed less to the T cell activation pathway (**A**,**B**). The Parkinson disease pathway to be more abundant in the non-severe form of the disease (**C**). The angiogenesis pathway was significantly enriched in patients with severe forms of sepsis (i.e., septic shock) (**D**).

**Figure 5 biomedicines-11-00778-f005:**
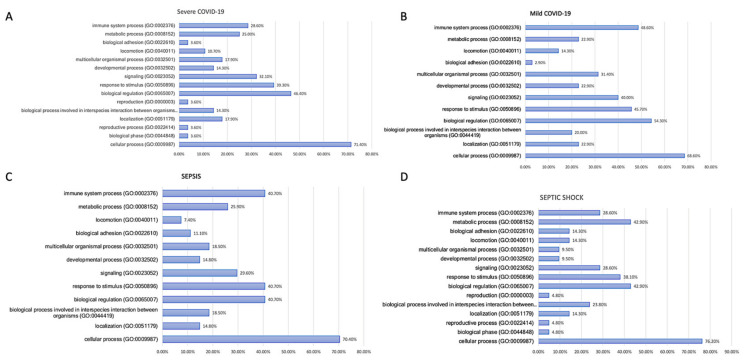
Biological process of the most commonly expressed genes in different cell subtypes across patients. Individually, the unique, most commonly expressed genes in COVID-19 and bacterial sepsis patients with mild and severe disease manifestations, found in Appendix A, were submitted to the PANTHER software. A biological process involvement analysis of the genes was returned by the software and was later coded into bar charts. The immune system and multicellular organismal processes were found enriched in the mild COVID-19 cohort (**A**,**B**) and in the non-severe sepsis cohort (**C**), indicating that both viral and bacterial induced sepsis are similar with respect to their biological processes. Septic shock patients demonstrated a higher upregulation of the metabolic pathway when compared to both severe COVID-19 patients and those with a mild form of the disease (**D**).

**Figure 6 biomedicines-11-00778-f006:**
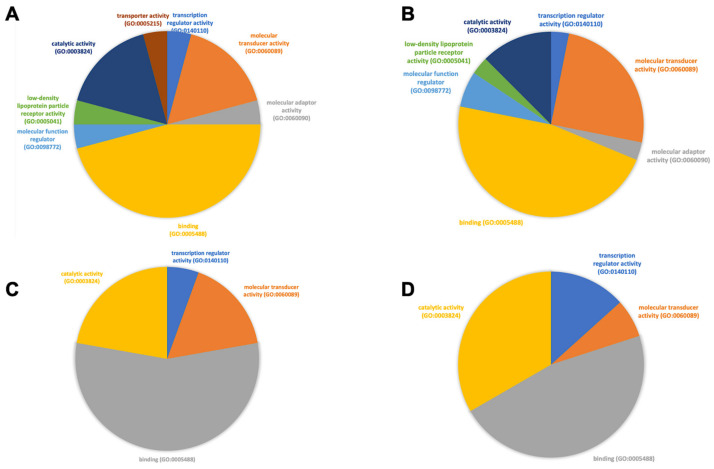
Molecular function of the most commonly expressed genes in different cell subtypes across patients. Individually, the unique, most commonly expressed genes in COVID-19 and bacterial sepsis patients with mild and severe disease manifestations, found in Appendix A, were submitted to the PANTHER software. A molecular function involvement analysis of the genes was returned by the software and was later coded into pie charts. Less molecular functions were performed by the genes involved in the sepsis cohort (**C**), i.e., catalytic activity, transcription regulator activity, molecular transducer activity, and binding. The catalytic activity and binding functions were more abundant in patients with severe sepsis (i.e., septic shock, **D**), while the remaining functions were higher in patients with non-severe sepsis. There was a higher enrichment of functions associated with the genes in the COVID-19 patients (**A**,**B**).

**Table 1 biomedicines-11-00778-t001:** Patient clinical characteristics and parameters.

DIAGNOSIS	SEPSIS	SEPTIC SHOCK	SEVERE COVID-19	MILD COVID-19
Age	57.25	65.00	65.25	76.33
male Sex (%)	75	75	50	66
Weight (kg)	86.25	100.00	82.35	76.27
Height (cm)	183.63	159.00	170.50	171.00
Surgical Admission	0.25	0.50	0.25	0.33
TEMP (°C)	37.88	39.70	37.30	37.47
RR (bpm)	29	36	30	22
GCS	3	3	7	15
PaO_2_/FiO_2_ (worst)	15	17	15	29
LACTATE (mmol/L)	2.3	2.86	2.43	2.77
Noradrenaline dose day of sample, min-max (mean), mcg/kg/min	0–0.667(0.149)	0.02–0.353(0.115)	0–0.169(0.031)	0.00
CRRT (%)	25	25	50	0
CREATININE (mmol/L)	152.50	182.00	92.50	95.67
egfr (ml/min/1.73 m^2^)	19.67	33.00	59.50	52.00
BILIRUBIN (µmol/L)	21	18	12	17
HB (g/dL)	10.33	7.10	12.35	11.67
PLT (10^9^/L)	305.00	717.00	346.75	332.00
WCC (10^9^/L)	28.83	18.60	12.83	8.70
APTT (seconds)	42.73	30.20	54.50	31.67
CRP (mg/L)	229.70	341.89	74.35	96.80
PCT (ng/mL)	0.32	0.48	0.21	
FERRITIN (µg/L)	15,207.40		1469.00	763.93
DDIMER (mg/L FEU)	1378.00		3386.75	780.00
NEUTS (10^9^/L)	25.43	14.90	10.85	5.17
LYMPHOCYTES (10^9^/L)	0.38	0.80	0.98	0.77
APACHE	37.00	41.00	21.00	9.00
SOFA	9.75	12.75	7.00	1.33
Hospital Length of Stay (days)	124	149	17	31
Survivors (%)	75	75	75	66
ICU LOS (days)	24.25	38.5	24.5	0

CRP = C-reactive protein; ICU = intensive care unit; PCT = procalcitonin; ICU LOS = ICU length of stay; GCS = Glasgow coma scale/score; CRRT = continuous renal replacement therapy.

## Data Availability

The data that support the findings of this study are available on request from the corresponding author. The data are not publicly available due to privacy or ethical restrictions.

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
