# Peer review of "Parallel Dysregulated Immune Response in Severe Forms of COVID-19 and Bacterial Sepsis via Single-Cell Transcriptome Sequencing"

_biomedicines, 2023, doi:10.3390/biomedicines11030778_

Round 1

Reviewer 1 Report

Thanks for the opportunity to review this paper regarding the cell-type-specific dysregulated immune response and immune silence in critically ill and non-severe patients with viral and bacterial-induced sepsis.

Authors try to delineate the differences and parallelism of patients with sepsis and COVID-19 with different degrees of severity and it is a very interesting and enough original topic.

This study could be of interest and of clinical relevance for the readers of Biomedicines.

The paper is clearly written.

Specific comments:

1. Study limitations are not clearly and extensively discussed in the text: I would recommend integrating this part into the discussion section.

2. For readers it would be interesting a table with clinical information about patients.

3. Finally, it would be interesting to discuss the possible therapeutic impact of authors’ findings.

Reviewer 2 Report

Review Biomedicines -2160973

The authors address a very interesting topic. However, the paper has several biases.

1. Patients' eGFR is not indicated in "materials and methods". We speak of "renal failure" see line 188 in a generic way. In fact, the immune system and lymphocyte subpopulations are linked to the eGFR level. Therefore I recommend reading and citing in the references the following two papers which refer to patients in hemodialytic treatment, but are useful for the relationship between natural immunity and renal disease:

- Lacquaniti A, Campo S, Falliti G, Caruso D, Gargano R, Giunta E, Monardo P.

Free Light Chains, High Mobility Group Box 1, and Mortality in Hemodialysis

Patients. J Clin Med. 2022 Nov 23;11(23):6904. doi: 10.3390/jcm11236904. PMID:

36498479; PMCID: PMC9739300.

- Campo S, Lacquaniti A, Trombetta D, Smeriglio A, Monardo P. Immune System

Dysfunction and Inflammation in Hemodialysis Patients: Two Sides of the Same

Coin. J Clin Med. 2022 Jun 28;11(13):3759. doi: 10.3390/jcm11133759. PMID:

35807042; PMCID: PMC9267256.

2. The paper talks about patients in septic shock, but the setting of the patients is not clear. Septic shock is linked to AKI and respiratory failure. In such cases the authors need to clarify the characteristics of patients in septic shock.

3. The authors must insert a table with the hematochemical, hemodynamic parameters. Furthermore, the antibiotic therapy (precise and not only indicated) and other drugs used (eg amines) must be clarified.

Authors must make the required revisions before publication

Round 2

Reviewer 2 Report

The authors made the required revisions

Author Response

We have reviewed the editors comments and ammended the manuscript ( title and abstract accordingly)